# Characterization of Residual Stresses and Grain Structure in Hot Forging of GH4169

**Zibiao Wang [1,2], Guangsha Hou [1], Yang Zhao [3], Jianfei Sun [1,\*], Jiangzhen Guo [1,4,\*] and Wuyi Chen [1]**

1 School of Mechanical Engineering and Automation, Beihang University, Beijing 100191, China; scientistbiao@163.com (Z.W.); mansionhou@163.com (G.H.); wychen@buaa.edu.cn (W.C.)
2 School of Mechanical Engineering, Tsinghua University, Beijing 100084, China
3 Beijing Scienkong Technology Co., Ltd., Beijing 100010, China; 18382414376@163.com
4 School of Engineering Medicine, Beihang University, Beijing 100191, China
\* Correspondence: sjf@buaa.edu.cn (J.S.); jzguo@buaa.edu.cn (J.G.)

**Abstract:** Residual stresses (RS) in hot forging severely degrade the machining accuracy and stability of super alloy parts. This is the main reason for deformation during subsequent mechanical machining. RS need recognition, as well as the microstructure and properties achieved by forging. In this study, a simulation and experimental research on the single-pass compression of GH4169 are presented. RS variations with forging temperature, loading speed, and cooling speed are established by finite element (FE) simulation. Based on the FE results, an experiment is conducted at a temperature of 1020 °C, loading speed of 25 mm/s, and press amount of 16 mm, immediately followed by water cooling. A new layer-stripping method is put forward for the high-efficiency measurement and correction of interior RS. Compared with the traditional strain gauge layer-stripping method, the measurement efficiency of the new layer-stripping method is increased by 10 times. Meanwhile, grain photographs are collected and grain size evolution is summarized; thus, the RS is characterized and evaluated from the angle of grains. It is demonstrated that the RS level rises with the increase in forging temperature, loading speed, and cooling speed, while the cooling method influences both the stress value and distribution. Compressive RS changes to tensile, while the average grain size reduces from the surfaces to the center. In the compressive regions, stress values share the same rules as grain size, while, in the tensile regions, they are contrary. The RS levels are divided according to the grain degree standard. According to the residual stress and grain distribution law of the blank, the optimal position of the part in the blank can be determined. Compared with the center position of the part in the blank, the residual stress of the part is reduced by 70%. The results provide useful strategies for the better design of forging technology, qualification examinations, and subsequent mechanical machining.

**Keywords:** residual stresses; grain size; hot forging; super alloy



## 1. Introduction

Nickel-based super alloy GH4169 (GB/T14992) has achieved widespread use in the aerospace industry due to the specific high strength and antioxidant property at high temperatures. It is the critical material in the manufacturing of thermal parts of aeroengines, such as turbine disks, blades, and cartridge receivers [1]. These parts are usually employed in extreme situations; thus, there are strict requirements regarding their qualities and mechanical properties. Forging is the main approach to obtain such semi-finished products of aeroengine hot sections. The GH4169 super alloy is sensitive to hot forging as problems such as coarse grains can easily occur. However, research by Brand et al. [2] suggested that the grain coarseness of GH4169 cannot be eliminated by recrystallization during heat treatment. The grain size must be controlled during hot forming; therefore, the forging technology becomes significant and is associated with many challenges. After the thermal

and mechanical cycles of heating, holding, and deforming [3], GH4169 forging achieves the desired microstructure and ultrasonic qualities, which guarantee their behavior in-service. At the same time, the coarsening behavior of the strengthening phase does not occur under the guarantee of immediate quenching [4]. Hot forging is a dynamic event with complex nonuniform temperature distribution as well as elastic and plastic deformation [5], but, unfortunately, the control of simultaneous RS with large gradients is difficult to realize. Research [6,7] shows that the magnitude of RS in GH4169 forged disks after quenching is over 400 MPa, and it is difficult to release fully by annealing and aging. RS release is a potential danger as it may degrade the accuracy and stability of the parts. It also causes severe machining deformation with the removal of materials. Overall, qualified forgings should meet the following requirements: uniform and low-level RS, as well as homogeneous and refined grains.

The parameters of hot forging are important as they directly affect the behavior of RS and grains. Wallis et al. [8] analyzed the influences of forging temperature and cooling rate on RS by the finite element method (FEM). They carried out the work in two different conditions, oil cooling and air cooling, and pointed out that the cooling rate is the key factor for microstructure stability and RS control. Cihak et al. [9] collected data on the interior RS of forged turbine disks produced from Inconel 718 with the neutron diffraction method. They compared the results under water-cooling conditions with those achieved by FEM and concluded that the forging parameters will change the state of RS. Marcelin [10] optimized the cooling conditions of hot-rolled beams with genetic algorithms and neural networks, and a cooling regime was found to minimize the RS. Ameli et al. [11] revealed the effects of parameters on RS, including the geometries of the workpiece and die, workpiece motions, and the percentage of deformation in cold radial forging. Chen et al. [12] performed hot compression tests of GH4169 and summarized the effects of the deformation degree, strain rate, and temperature on dynamic recrystallized grains. Azarbarmas et al. [13] revealed the dynamic recrystallization (DRX) of Inconel 718 with optical methods and EBSD images. They also proposed a constitutive equation for modeling and predicting the peak stress. The model identified that values of RS related only to the strain rate and temperature. Loyda et al. [14] used a numerical model to evaluate grain size during rotary forging with three temperatures. The study explored the influence of forging on microstructure evolution; thus, a homogenous and refined grain size could be obtained. Li et al. [15] set up a calculation model of RS during hot stamping that could precisely describe the stress distribution. Xu et al. [16] focused their research on forged plate parts of Inconel 718 and proved that changes in RS in the axial direction were similar to the dislocation density. They compared the stress results under different forging methods with both neutron diffraction and X-ray diffraction. They also showed differences in measuring methods. Therefore, the forging temperature, loading speed, and cooling speed are critical for hot forging and exert various degrees of impact on the value and distribution of RS.

In fact, the mechanical properties of GH4169 are sensitive to grain size, which is altered by the forging temperature and strain rate [17]. Many researchers have discussed the microstructure evolution of GH4169 during hot forming from the perspectives of theoretical analysis and numerical simulation. They concentrated on the specific forming process and proposed ideal prediction models. Ma et al. [18] predicted the variations in flow stress and grain size in the multistage heavy forging of Inconel 718. They built microstructure models considering both the strain hardening effect and dynamic softening effect. Such models can be used to optimize the processing parameters to obtain the desired parts. Toro et al. [19] designed a fuzzy inference system to predict the grain size of Inconel 718 after upset forging considering the reduction rate and temperature. Berti et al. [20] developed a set of equations to predict microstructure and geometry evolution. They provided guidance for the rolling of parts on different scales. Wang et al. [21] developed compression tests of Inconel 718 cylindrical specimens with different types of grain size distribution. Results showed that a homogeneous microstructure with fine grains could be obtained at a low temperature and slow strain rates. Zhu et al. [22] studied the grain

structure evolution during the forging of GH4169 by FE simulation. They proposed a DRX model and pointed out that the evolution mechanism under several stages is different. Investigations should be carried out based on a specific process. Tang et al. [23] presented an internal state variable material model that can be used to predict the flow behavior during the dynamic regime. They also modeled the microstructure evolution of nickel-based super alloys based on the experimental results [24]. A set of unified constitutive equations were proposed and further used to analyze the microstructure evolution with the overall flow stress. Results indicated that thermal stress varied only with deformation temperature and strain rate, while the stress distribution changed with the grain size evolution. Zhbankov et al. [25] proposed new forging schemes with rational parameters, which provide uniform distribution of grain size. Due to the fact that there is no phase transformation in nickel-based super alloy during high-temperature deformation, DRX was the only way to control the grain size [26]. The synthetic study of grain size and RS control can be carried out during single forging.

In this paper, simulations and experimental research on single-pass compression are presented with a press amount of 16 mm. At first, RS variations with forging temperature, loading speed, and cooling speed are established by FE simulations. The findings provide further support for optimizing the forging parameters to reduce stress levels. Then, a compressing experiment is conducted at a temperature of 1020 °C and loading speed of 25 mm/s. This is immediately followed by water cooling. Based on the experiment, RS tests and strain photographs are assessed in the forging direction. A new layer-stripping method is put forward to obtain the RS distributions. The method is unique as it not only improves the measuring range and resolution, but it also compensates for the error induced by layer stripping. Finally, the simulation and test results of RS are compared and analyzed according to the effective strain and temperature chart of the part. The grain size evolution is summarized too. Combining the trends of RS and grain size, the optimum workpiece machining sequence and position can be realized.

## 2. Materials and Methods

### 2.1. Details of Forging Process

Table 1 shows the chemical composition and mechanical properties at room temperature of the studied GH4169 super alloy. Specimens with the geometric dimensions of ⌀36 mm × 110 mm were machined by cold drawing, solid solution, and polish treatment. The microstructures of the specimens before compression are shown in Figure 1.

**Table 1.** Chemical composition and mechanical properties of GH4169.

| Elements | C | Cr | Co | Al | Ti | Nb | Ni |
|---|---|---|---|---|---|---|---|
| wt% | ≪ 0.08 | 17.0~21.0 | ≤ 1.0 | 0.30~0.70 | 0.75~1.15 | 4.75~5.50 | Bal. |
| Mechanical properties | | | | Temperature (20 °C) | | | |
| Tensile strength $\sigma_b$(MPa) | | | | 1372 | | | |
| Elongation δ(%) | | | | 14.6 | | | |
| Coefficient of thermal expansion ($10^{-6}$/°C) | | | | 13.2 | | | |
| Thermal conductivity (W/(m °C)) | | | | 13.4 | | | |

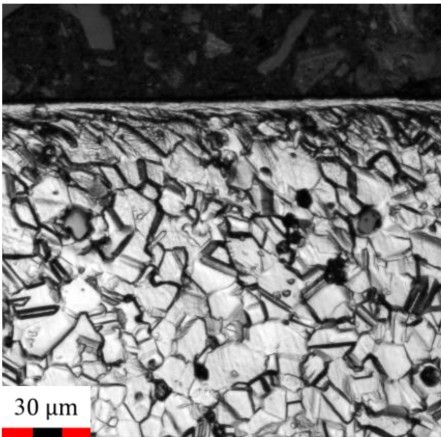

**Figure 1.** Microstructure of the specimens before compression.

The single-pass compression was performed in Forging Machine DP-C41L13 at a forging temperature of 1020 °C, loading speed of 25 mm/s, and press amount of 16 mm. The depth of the compressed specimen was 20 mm and it was water-cooled immediately. The specimen shapes before and after forging are shown in Figure 2.

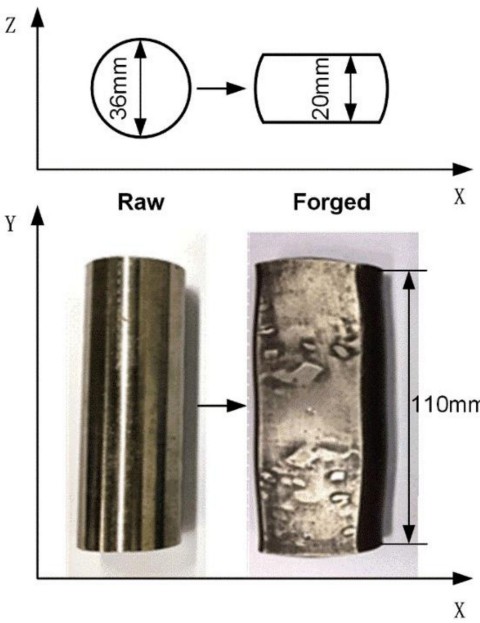

**Figure 2.** Specimen shapes before and after forging.

### 2.2. Measurement Method of Residual Stresses

A measurement method of RS with high accuracy and efficiency is essential in experimental investigations. The blank initial residual stress measuring range is required to be more than 10 mm. The layer-stripping method is a traditional method to measure the residual stress profile of the blank. The traditional layer-stripping method to measure the residual stress needs to remove a layer of material and attach a strain gauge, which takes around 1.5 h. Moreover, the traditional layer-stripping method is inevitably limited by material removal, including the low precision and resolution caused by excessive removal thickness, and the interferences of stripping-induced RS when the removal thickness is small. The traditional layer-stripping method obtains the residual stress through the strain gauge, and the measurement accuracy of the residual stress depends on the measurement accuracy of the strain gauge, which is generally around 50 MPa [27,28]. The stress state is difficult to precisely describe. The proposed new layer-stripping method combines the

PRISM and layer-stripping method. The PRISM (Stresstech Oy, Jyväskylä, Finland) is a specialized piece of equipment for RS measurement combining the laser interference technique and hole drilling method, as pictured in Figure 3. Its maximum measurement depth was 2 mm. The measurement accuracy was 7 MPa. The RS depth profile was measured via incremental drilling. First, a small hole was drilled into the part's surface, the RS relaxation resulting in a tiny surface deformation. Afterwards, this deformation was measured by electronic speckle pattern interferometry [29–32]. The proposed new method measures residual stress for around 5–10 min at a time. Furthermore, measuring results will be corrected considering the stress redistribution caused by layer stripping.

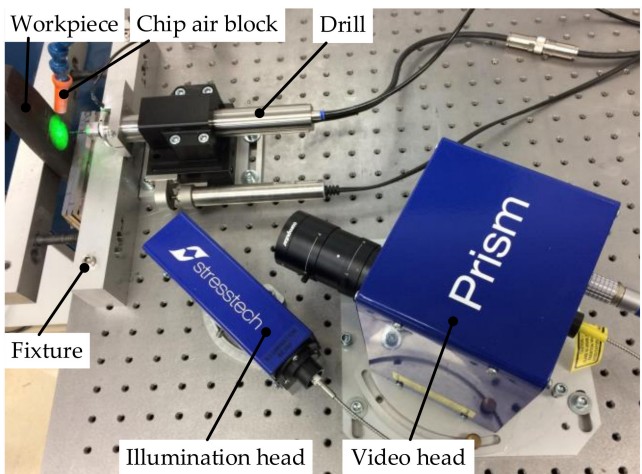

**Figure 3.** RS measurement arrangement (PRISM).

　　　Surface RS within 2 mm can be measured precisely every 0.2 mm by the PRISM and the first measured layer of 2 mm will then be removed. RS on the second layer (new surface layer) of 2 mm will be measured and then the second measured layer is removed. Removing the measured layer successively, the following layer can then be measured, and the cycle makes the deep layers measurable. It can be seen from the forging simulation results that, during the forging process, the boundary conditions, such as the load, restraint, and temperature of the part, are all symmetrical around the middle of the part, and the stress–strain distribution state of the part after forging is also symmetrically distributed along the symmetry plane. Therefore, only half of the specimen is depicted [33]. The half of the specimen, which is 10 mm in thickness, is divided into 5 layers uniformly and measured. The layer-stripping process is performed by wire-electrode cutting. The measuring process and regions are shown in Figure 4, where region C occupies the geometric center of the specimen.

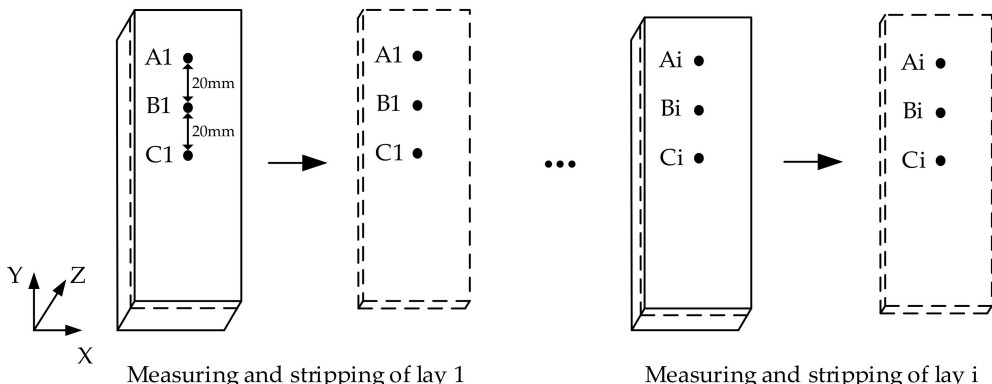

**Figure 4.** Measurement method and specimens with measuring positions.

　　　Figure 5 shows the stress redistribution and corresponding deformation after removing the first layer. Suppose that the material thickness is $h_0$ and it is evenly divided into

$n$ layers. Symbol t represents the thickness of each layer. Symbol $h_i$ equals the thickness of the rest of the material after layer i is removed. Symbol $l$ represents the dimensions of length and width. Each time a layer is removed, it will produce new stresses in the rest of the material to reach a stress balance. Symbols $\sigma_{xi}$ and $\sigma_{yi}$ respectively represent the blank initial residual stress in the directions of X and Y in layer i before removal; symbols $M_{xi}$ and $M_{yi}$ represent the corresponding moments. Symbols $S_{xi}$ and $S_{yi}$ represent the measured residual stress in layer i with PRISM. Symbols $\sigma_{x(i,j)}$ and $\sigma_{y(i,j)}$ represent the new stress balance in layer j caused by the removal of layer i. Symbols $\sigma_{xi}'$ and $\sigma_{yi}'$ represent the resultant stress before the rest of the material of the part reaches the new stress balance again, after layer i is removed, and symbols $M_{xi}'$ and $M_{yi}'$ represent the corresponding moments.

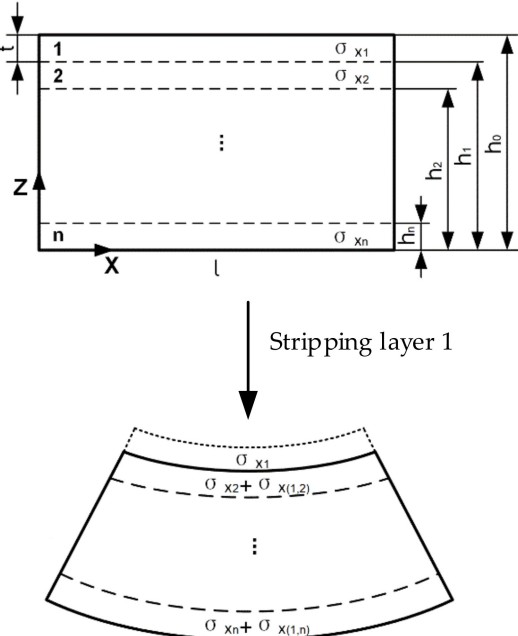

**Figure 5.** Stress redistribution and corresponding deformation.

Taking the first layer stripping as an example, there is a linear correlativity between the new stress balance and the thickness of the rest of the material.

$$\begin{cases} \sigma_{x1}(z) = \frac{\sigma_{x(1,2)} - \sigma_{x(1,n)}}{h_1} z + \sigma_{x(1,n)} \\ \sigma_{y1}(z) = \frac{\sigma_{y(1,2)} - \sigma_{y(1,n)}}{h_1} z + \sigma_{y(1,n)} \end{cases} \tag{1}$$

where $\sigma_{x1}(z)$ and $\sigma_{y1}(z)$ represent the new stress balance after layer i is removed. Thus, the corresponding net moments $M_{x1}'$ and $M_{y1}'$ can be described as

$$\begin{cases} M_{x1}' = \int_0^{h_1} \sigma_{x1}(z) \cdot l \cdot z d_z = \frac{1}{3} \cdot \frac{\sigma_{x(1,2)} - \sigma_{x(1,n)}}{h_1} \cdot l \cdot h_1{}^3 + \frac{1}{2} \cdot \sigma_{x(1,n)} \cdot l \cdot h_1{}^2 \\ M_{y1}' = \int_0^{h_1} \sigma_{y1}(z) \cdot l \cdot z d_z = \frac{1}{3} \cdot \frac{\sigma_{y(1,2)} - \sigma_{y(1,n)}}{h_1} \cdot l \cdot h_1{}^3 + \frac{1}{2} \cdot \sigma_{y(1,n)} \cdot l \cdot h_1{}^2 \end{cases} \tag{2}$$

Considering the equilibrium conditions of stress and moment, there are equality relations before and after removal.

$$
\begin{cases}
\sigma_{x1} = -\sum_{i=2}^{n} \sigma_{xi} = \sigma_{x1}' \\
\sigma_{y1} = -\sum_{i=2}^{n} \sigma_{yi} = \sigma_{y1}' \\
M_{x1} = -\sum_{i=2}^{n} M_{xi} = M_{x1}' \\
M_{y1} = -\sum_{i=2}^{n} M_{yi} = M_{y1}'
\end{cases}
\tag{3}
$$

where the elements can be calculated as

$$
\begin{cases}
M_{x1} = \sigma_{x1} \cdot l \cdot t \left(h_0 - \frac{t}{2}\right) \\
M_{y1} = \sigma_{y1} \cdot l \cdot t \left(h_0 - \frac{t}{2}\right)
\end{cases}
\tag{4}
$$

$$
\begin{cases}
\sigma_{x1}' = \frac{h_1}{2t} \left(\sigma_{x(1,2)} + \sigma_{x(1,n)}\right) \\
\sigma_{y1}' = \frac{h_1}{2t} \left(\sigma_{y(1,2)} + \sigma_{y(1,n)}\right)
\end{cases}
\tag{5}
$$

According to Equations (3)–(5), $\sigma_{x(1,2)}$ and $\sigma_{y(1,2)}$ can be calculated as

$$
\begin{cases}
\sigma_{x(1,2)} = -\frac{2t}{h_1^2} \cdot \left[3\left(h_0 - \frac{t}{2}\right) - h_1\right] \cdot \sigma_{x1} \\
\sigma_{y(1,2)} = -\frac{2t}{h_1^2} \cdot \left[3\left(h_0 - \frac{t}{2}\right) - h_1\right] \cdot \sigma_{y1}
\end{cases}
\tag{6}
$$

where $\sigma_{x1} = S_{x1}$ and $\sigma_{y1} = S_{y1}$ can be measured by PRISM. The initial RS in layer 2 can be corrected as

$$
\begin{cases}
\sigma_{x2} = S_{x2} - \sigma_{x(1,2)} \\
\sigma_{y2} = S_{y2} - \sigma_{y(1,2)}
\end{cases}
\tag{7}
$$

Then, layer 2 will be removed, and the calculated results of $\sigma_{x2}$ and $\sigma_{y2}$ can be used as the known condition. Similarly, the results obtained by PRISM can be corrected successively. It is noted that the new stress balances overlap with the increase in layer-stripping times. For convenience of calculation, Equation (1) can be rewritten as

$$
\begin{cases}
\sigma_{x(1,j)} = \frac{\sigma_{x(1,2)} - \sigma_{x(1,n)}}{h_1} \cdot \left(h_0 - j \cdot t + \frac{t}{2}\right) + \sigma_{x(1,n)} \\
\sigma_{y(1,j)} = \frac{\sigma_{y(1,2)} - \sigma_{y(1,n)}}{h_1} \cdot \left(h_0 - j \cdot t + \frac{t}{2}\right) + \sigma_{y(1,n)}
\end{cases}
\tag{8}
$$

where $\sigma_{x(1,n)}$ and $\sigma_{y(1,n)}$ can be calculated based on Equation (5):

$$
\begin{cases}
\sigma_{x(1,n)} = \frac{6t}{h_1^2} \left(h_0 - \frac{t}{2}\right) \cdot \sigma_{x1} \\
\sigma_{y(1,n)} = \frac{6t}{h_1^2} \left(h_0 - \frac{t}{2}\right) \cdot \sigma_{y1}
\end{cases}
\tag{9}
$$

This can be adapted to the general cases as follows:

$$
\begin{cases}
\sigma_{x(i,j)} = \frac{\sigma_{x(i,i+1)} - \sigma_{x(i,n)}}{h_i} \cdot \left(h_0 - j \cdot t + \frac{t}{2}\right) + \sigma_{x(i,n)} \\
\sigma_{y(i,j)} = \frac{\sigma_{y(i,i+1)} - \sigma_{y(i,n)}}{h_i} \cdot \left(h_0 - j \cdot t + \frac{t}{2}\right) + \sigma_{y(i,n)}
\end{cases}
\tag{10}
$$

$$
\begin{cases}
\sigma_{xi} = S_{xi} - \sum_{m=1}^{i-1} \sigma_{x(m,i)} \\
\sigma_{yi} = S_{yi} - \sum_{m=1}^{i-1} \sigma_{y(m,i)}
\end{cases}
\tag{11}
$$

The traditional layer-stripping method uses the strain gauge to monitor deformation and further compute the RS. Compared with the traditional strain gauge layer-stripping method, the measurement efficiency of the new layer-stripping method is increased by 10 times. To further confirm the accuracy of PRISM, deformation of the first removed layer is calculated by the measuring results based on the theory reported by Wang et al. [34]. It can be seen in Figure 6 that the analytical results match well with the reality. The maximum values of deformation are 1.16 mm and 0.93 mm, respectively, and the error is around 20%. It is confirmed that the PRISM is reliable.

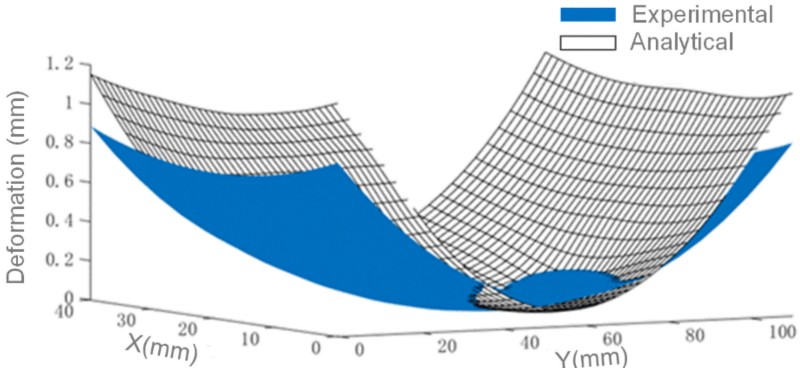

**Figure 6.** Experimental and analytical results of deformation in layer 1.

## 3. Simulations of Residual Stresses with Different Parameters

The finite element software used for the forging simulation was DEFORM. The forging simulation model is shown in in Figure 7. The friction coefficient between the part and the mold was 0.2 [35]. Forging dies were set at room temperature. The Young's modulus of GH4169 was 204 GPa and Poisson's ratio was 0.30. Considering the symmetrical conditions of geometry, loads, and constraint, the simulation model contained only one-eighth of the specimen to reduce the running time and memory space. The element style was tetrahedrons with high versatility and the total amount was 54,000. The sparse solver and the Newton–Raphson function were adopted.

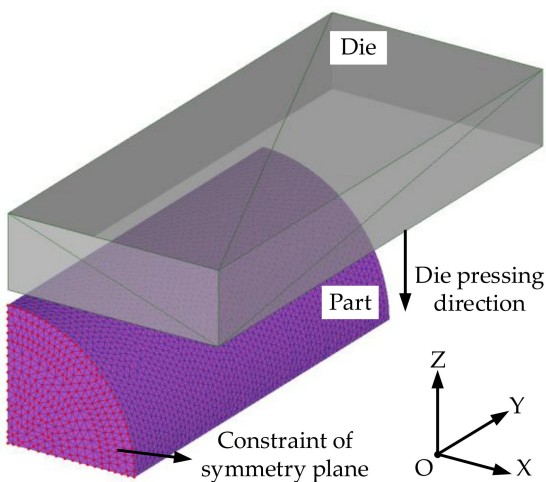

**Figure 7.** Forging simulation model.

The variables consisted of the cooling method, forging temperature, and loading speed. Firstly, three cooling methods, namely air cooling, oil cooling, and water cooling, were chosen to simulate the forged models obtained at the forging temperature of 1020 and loading speed of 25 mm/s. The convective heat transfer coefficient curves used in the FE simulations based on data in [36] are presented in Figure 8. Secondly, the loading speed of 25 mm/s was fixed, and models at different forging temperatures, 950 °C, 1020 °C,

1040 °C, and 1100 °C, were investigated. All the models were water-cooled immediately after compression. Thirdly, at the forging temperature of 1020 °C, models were compressed at a loading speed of 5 mm/s, 25 mm/s, 50 mm/s, and 100 mm/s, followed by water cooling. The effective stresses with various parameters were extracted and contrasted. The flow stress constitutive model used in the FE simulations is as follows:

$$\dot{\varepsilon} = A[sinh(\alpha\sigma)]^n exp\left(-\frac{Q}{RT}\right) \tag{12}$$

where $\dot{\varepsilon}$ denotes the strain rate ($s^{-1}$), $\sigma$ denotes the flow stress (MPa), and T denotes the deformation temperature (T). Q indicates the hot deformation activation energy (J/mol), and R indicates the gas constant (J/(mol·K)). A, n, and $\alpha$ are temperature-independent constants. They respectively represent the structural factor, the stress exponent, and the stress scale parameter. Table 2 lists the value of each parameter.

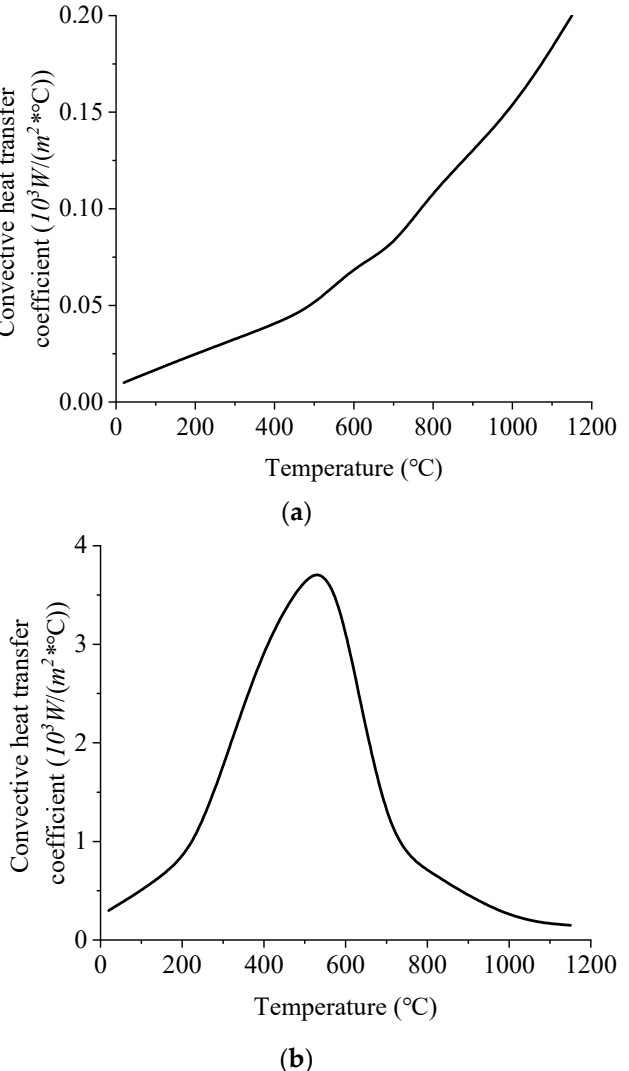

**Figure 8.** *Cont.*

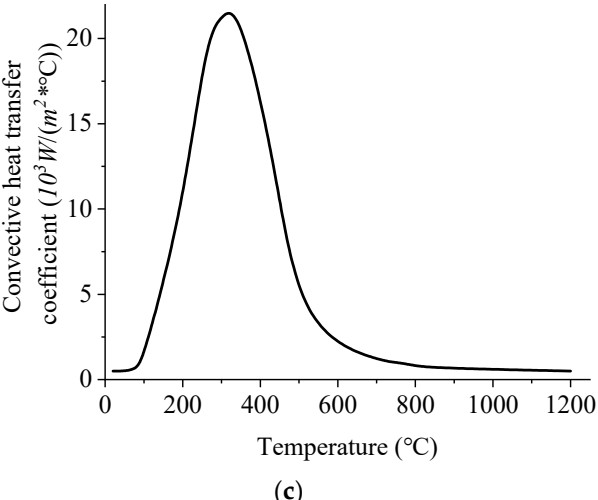

**(c)**

**Figure 8.** Convective heat transfer coefficient curves used in FE simulations: (**a**) air cooling; (**b**) oil cooling; (**c**) water cooling.

**Table 2.** Value of parameters in the constitutive model.

| Material | Q (J/mol) | R (J/(mol·K)) | A | n | α |
|----------|-----------|---------------|---|---|---|
| GH4169 | 413,118 | 8.314 | $4.51 \times 10^{16}$ | 5.05 | 0.0024 |

The simulation results of effective stress focused on the factors of cooling method, forging temperature, and loading speed are shown in Figures 9–11 respectively. There are protrusions at the edge of the part in the forging simulation. The main reasons for this phenomenon are as follows. Firstly, the upper and lower surfaces of the part are in contact with the forging tool during the forging process. The contact surface has a large friction force, so that the closer to the upper and lower surfaces, the greater the flow resistance of the material. However, the material in the middle of the part can flow freely, so drums form at the edges of the part. Secondly, the upper and lower surfaces of the part are in contact with the forging tools, and the heat transfer coefficient is large, which causes the temperature of the upper and lower surfaces of the part to cool rapidly. The lower the temperature is, the greater the deformation resistance is, so the closer the upper and lower surfaces are, the more difficult it is to deform.

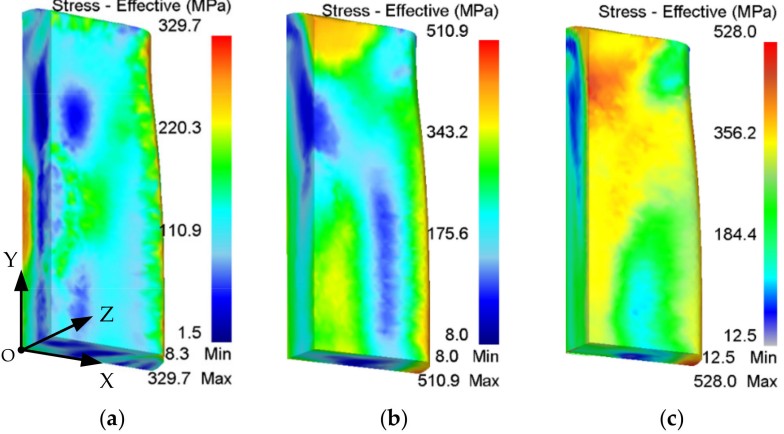

**Figure 9.** Simulation results of effective stress under T = 1020 °C and V = 25 mm/s. (**a**) air-cooling, (**b**) oil-cooling, (**c**) water-cooling.

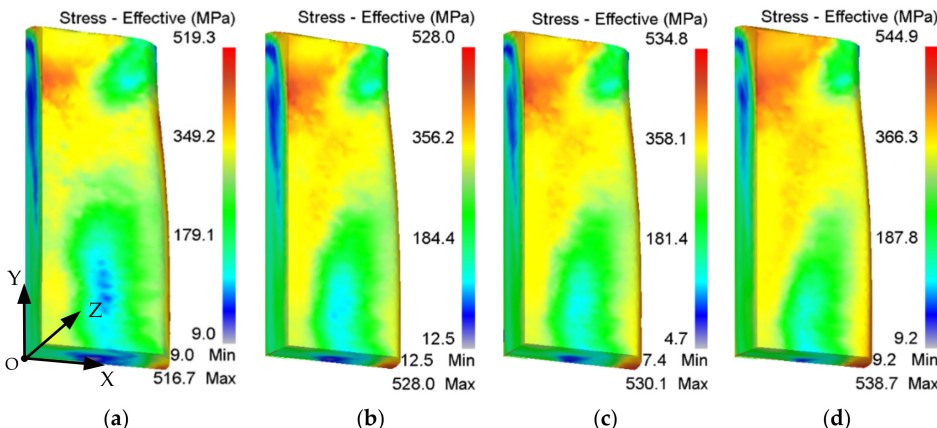

**Figure 10.** Simulation results of effective stress under V = 25 mm/s and water cooling. (**a**) T = 950 °C, (**b**) T = 1020 °C, (**c**) T = 1040 °C, (**d**) T = 1100 °C.

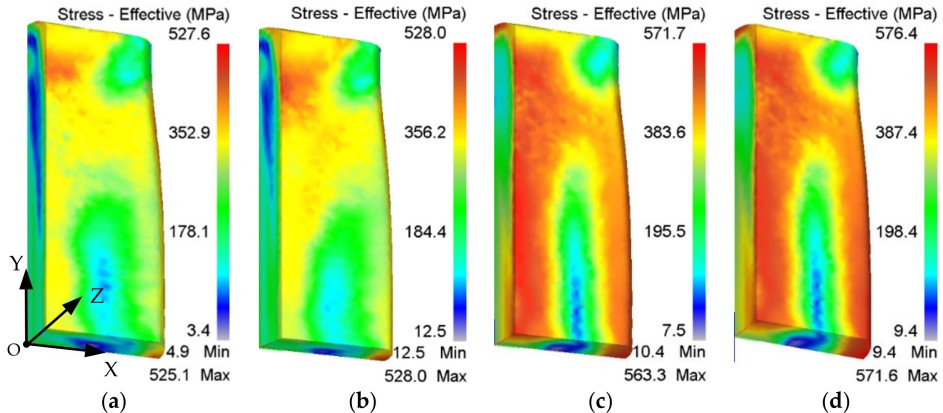

**Figure 11.** Simulation results of effective stress under T = 1020 °C and water cooling. (**a**) V = 5 mm/s, (**b**) V = 25 mm/s, (**c**) V = 50 mm/s, (**d**) V = 100 mm/s.

As seen in Figure 9, both the value and distribution of RS are strongly influenced by the cooling method. It is mainly caused by the differences in the convective heat transfer coefficient. Under the condition of air cooling, the coefficient continues to near-linearly increase with temperature, and the value appearing at the peak temperature of 1020 °C is approximately 150 W/$(m^2 \cdot °C)$. In this case, it maintains a relatively low stress level. The stresses are evenly distributed, with values lower than 200 MPa. Stresses over 300 MPa appear on the edge of the model and this has a minor effect on the whole part. The convective heat transfer coefficient increases nearly 100 times under the water-cooling condition; the maximum stress value increases significantly from 329.7 MPa to 528 Mpa, and the growth rate is around 60%. Regions with a stress level over 300 MPa are dominant. The enlarged convective heat transfer coefficient directly induces the increasing temperature gradient during cooling, which consequently changes the distribution of RS [37]. It is concluded that cooling methods with a small convective heat transfer coefficient are recommended to obtain forging pieces with low RS.

It can be seen from Figure 10 that discrepancies in RS induced by forging temperatures are mainly reflected in the values. At the temperature range of 950 °C to 1100 °C, the peak value changes from 519.3 MPa to 544.9 MPa. The raising temperature does increase the stress value as the amplification can be observed in the charts, but the extent is almost negligible.

Figure 11 shows the stress results with different loading speeds. The loading speed has a notable influence on the RS. Though the growth rate of the peak value is only approximately 9% (from 527.6 MPa to 576.4 MPa), a remarkable increment in stress appears in most of the regions. For example, the stress in the center of the model (region C) on the

surface is approximately 332 MPa at the loading speed of 5 mm/s. However, it increases to 531 MPa when the loading speed reaches 100 mm/s, with a growth rate of 60%.

It can be seen from Figure 12 that with the increase in the loading speed, the temperature of the part increases, so the residual stress generated during the cooling process is greater. This is because the faster the forging speed, the higher the strain rate of the part material. The higher the strain rate is, the greater the flow stress of the material during forging is, and the forging pressing amount under different forging speeds is equal. Therefore, the faster the forging pressing speed is, the more work the forging exerts on the workpiece, and the more heat is generated during the forging process. At the same time, since the forging pressing amount is the same, the faster the forging speed is, the shorter the contact time between the workpiece surface and the forging die is, the smaller the heat transfer from the workpiece to the forging die is, and the less heat the workpiece transfers to the air. Therefore, the temperature of the workpiece is higher after forging.

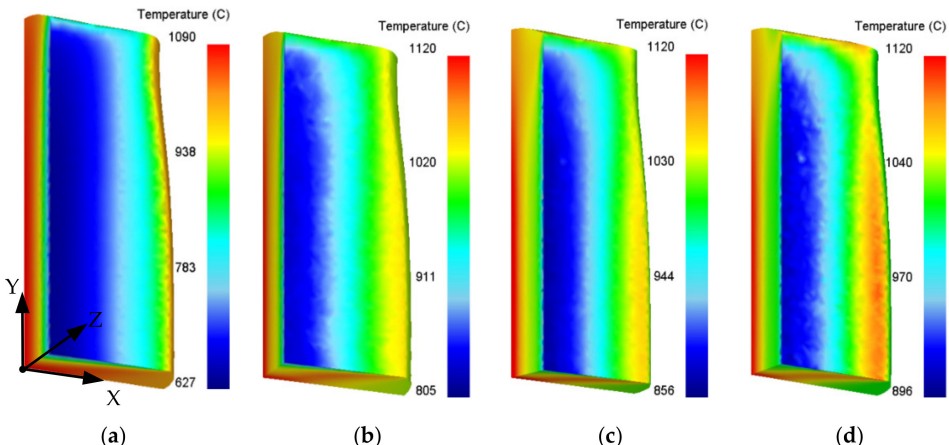

**Figure 12.** Simulation results of final forging temperature under T = 1020 °C. (**a**) V = 5 mm/s, (**b**) V = 25 mm/s, (**c**) V = 50 mm/s, (**d**) V = 100 mm/s.

## 4. Experimental Results and Discussion

### 4.1. Experimental Results of Residual Stresses

The residual stress of the forging blank is the main cause of machining deformation. The smaller the residual stress of the blank, the smaller the machining deformation caused by the release of residual stress in the machining process. In order to effectively control the generation of blank residual stress, this section describes the reduction of the residual stress in parts by optimizing the forging process parameters and the positions of the part in the blank. From the research results listed in the previous section, it can be seen that a low forging temperature, convective heat transfer coefficient, and loading speed are recommended to achieve stable parts with uniform and low-level RS. Then, the optimal positions of the part in the blank will be determined by analyzing the residual stress distribution law of the blank.

At the temperature of 1020 °C and loading speed of 25 mm/s followed by water cooling, the simulation and experimental results of RS were extracted and compared. Distributions of RS at a 20 mm depth in the Z direction in regions A~C are shown in Figure 13.

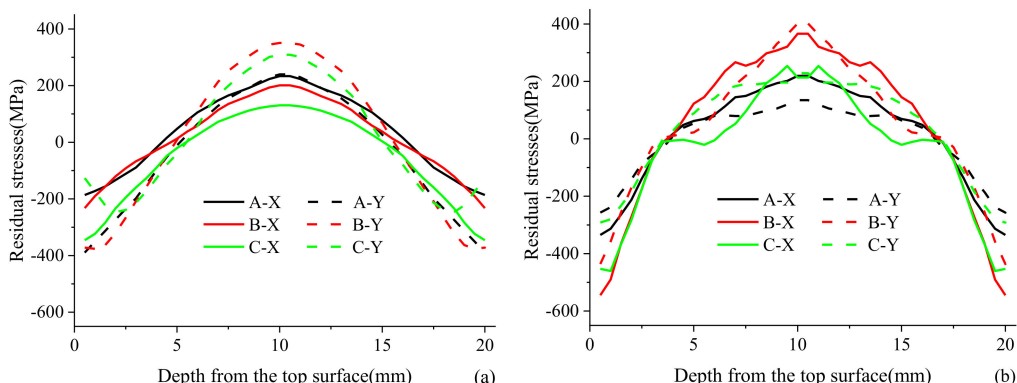

**Figure 13.** Distributions of residual stresses at depth: (**a**) simulation results; (**b**) measurement results.

The depths of 0 mm and 20 mm are the top and bottom surfaces of the specimen, respectively. It shows compressive stresses on the surfaces and tensile stresses in the center. The curves are nearly parabolic, as the peak values of compressive and tensile stresses appear on the surfaces and center, respectively. According to the stress distribution state, the best position of the part in the blank should be close to a quarter of the thickness of the blank from the surface. Compared with the center position of the part in the blank, the residual stress is reduced by 70%. Forging RS of GH4169 come from two main sources: the nonuniform plastic deformation during compression and the temperature gradients in cooling. The effective strain and temperature after compression are shown in Figure 14. Based on this, the RS distribution could be reasonably explained.

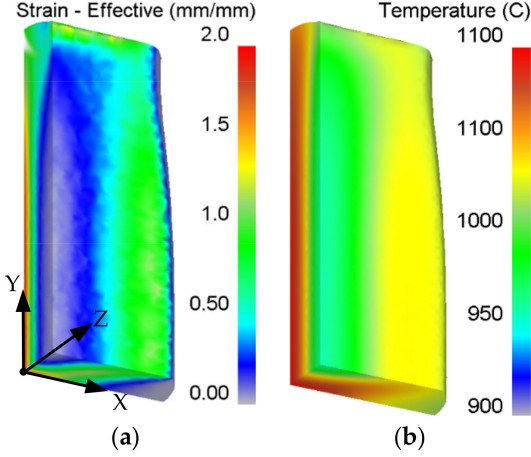

**Figure 14.** Simulation results of effective strain and temperature after compression. (**a**) Effectively strain, (**b**) Temperature.

The compression is completed in a short time, which results in severe elastoplastic deformation of the specimen under external loading. The deformation displacements show differences from the surfaces to the center, as do the plastic strains. It can be seen in Figure 14a that the compressive strain shows a decreasing trend from the surfaces to the center. After unloading, deformation recovery occurs towards the stable stress balance state and material interaction finally induces RS. The deformation release direction is opposite to the deformation direction produced by the forging. Due to the small elastoplastic deformation on the surfaces, the outer materials take the lead in finishing the recovery. The severely deformed inner materials continue recovering and the recovery is inevitably hindered by the outer materials.

The temperature gradient is the main reason that RS exist during cooling. Simultaneous cooling is challenging to realize during water cooling, as the outer materials come into direct contact with water and the temperature reduces sharply, while the inner temperature

drops slowly. The original temperature difference reaches 150 °C after compression and it will further increase during cooling. The temperature of the workpiece gradually increases from the surface to the center, and reaches the highest temperature at the center of the workpiece, as shown in Figure 14b. At the beginning of cooling, the material on the surface of the workpiece is rapidly cooled to a low temperature, and the outer material has reached a stable state. As the cooling progresses, the core temperature gradually decreases. Due to the effect of thermal expansion and contraction, the volume of the material inside the workpiece will gradually decrease. The reduction in the volume of the inner material causes the compression of the outer material, so that the outer material forms compressive stress. In contrast, the contraction of the inner material is hindered by the outer material, so the inner material produces tensile stress.

The measurement results are in firm agreement with the FE simulations in regions A~C, although the practical values of the compressive stresses are slightly larger. This further proves that the measurement method is reliable. The maximum values of compressive stresses appear on the surfaces with a magnitude around 300 MPa~500 MPa. The compressive layer is around 4 mm (one-fifth of the whole depth) individually from the top and bottom surfaces. Compressive values decrease with the increasing depth from the surface and there is a transition layer of 2 mm where it gradually converts to tensile. The thickness of the tensile RS layer is around 8 mm (two-fifths of the whole depth) lying in the center, and the curve shape resembles a mountain. The peak value of tensile stress located centrally reaches 200 MPa~400 MPa. The mountain-shaped RS are in good accordance with the temperature gradient. Based on the results, by adopting the following machining sequence symmetrically, the deformation can be effectively suppressed.

### 4.2. Experimental Results of Grain Structures

Grain structures were observed using the OLYMPUS OLS4100, a 3D measuring laser scanning confocal microscope. The sampling method and test positions are shown in Figure 15. Regions A~C correspond to those in the RS measurement. Three positions at different depths in each region were photographed, where position 1 is on the surface of the sample and position 3 is on the central plane.

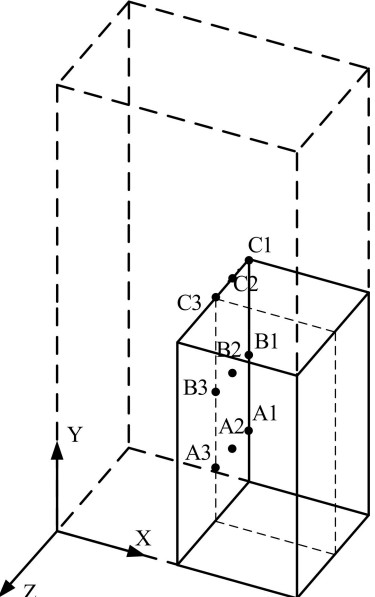

**Figure 15.** Metallographic specimen with test positions.

Photos of grain morphologies are shown in Figure 16, the subscript on the right side of each subfigure represents the corresponding measurement position of the grain structure

in Figure 15. Based on the photos, grain size was calculated and the distribution law is summarized in Figure 17.

Grain size in three different regions of the specimens followed a similar revolution regularity. It decreases with distance to the surface, which is contrary to that of the temperature. The grain size ranges from 35 μm to 50 μm in low-temperature areas on the surface, while refined grains of 3 μm~5 μm appear in the center. Grain growth occurs during heating and holding and it is hindered by the δ phase. The melting point of GH4169 is 1260~1320 °C, and the δ phase dissolves at around 1010 °C [38]. Therefore, the growth of grains is slow below 1010 °C, and it accelerates during the following heating cycle to 1020 °C. Adequate growth happens in the following holding stage and coarse grains measuring approximately 50 μm are obtained. Varying degrees of refinement occur with the heat compression, and this finally appears as differences in grain size. The position of the part in the blank should be chosen to be between one-quarter and three-quarters from the upper blank surface.

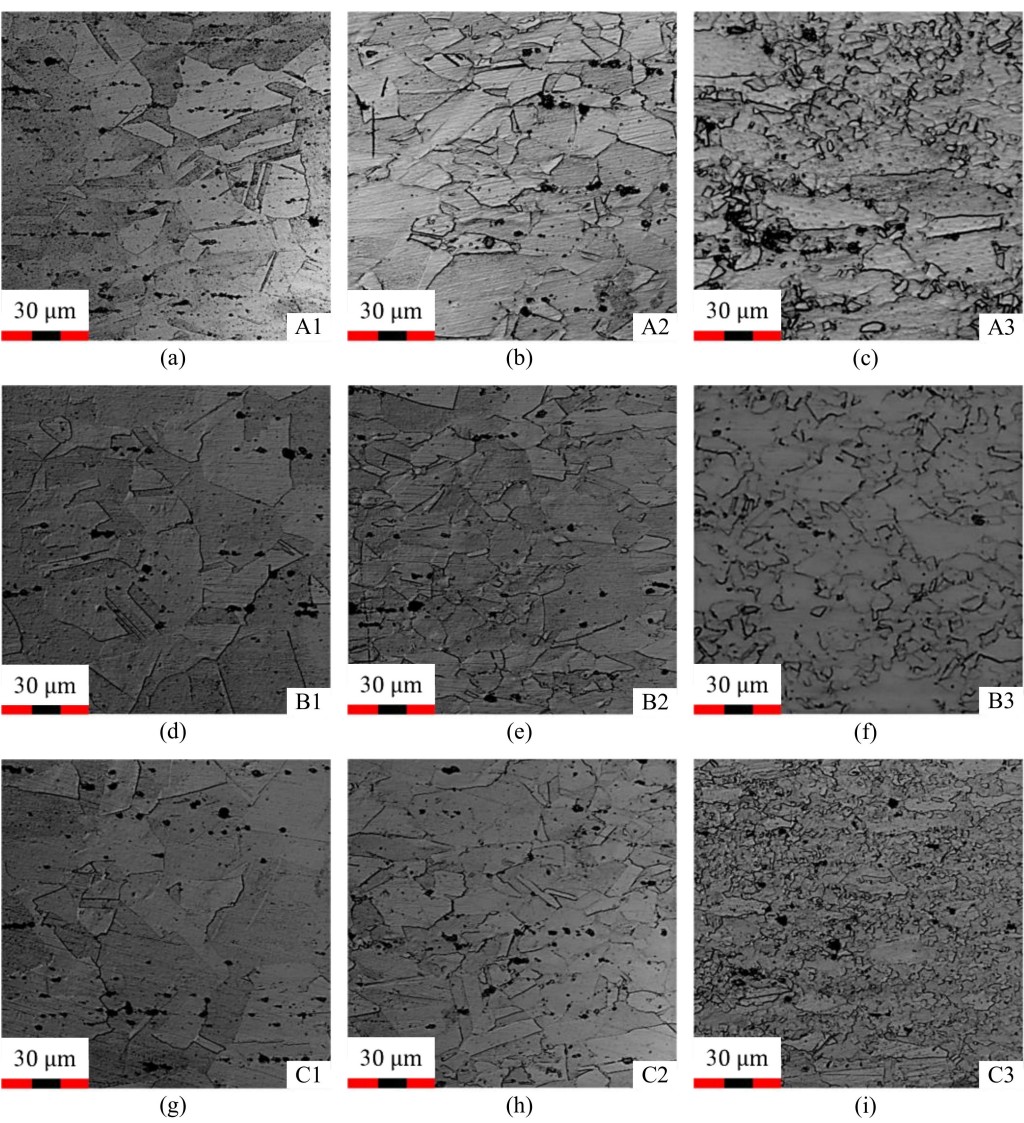

**Figure 16.** Grain morphologies at specified positions: (**a**) position A1; (**b**) position A2; (**c**) position A3; (**d**) position B1; (**e**) position B2; (**f**) position B3; (**g**) position C1; (**h**) position C2; (**i**) position C3.

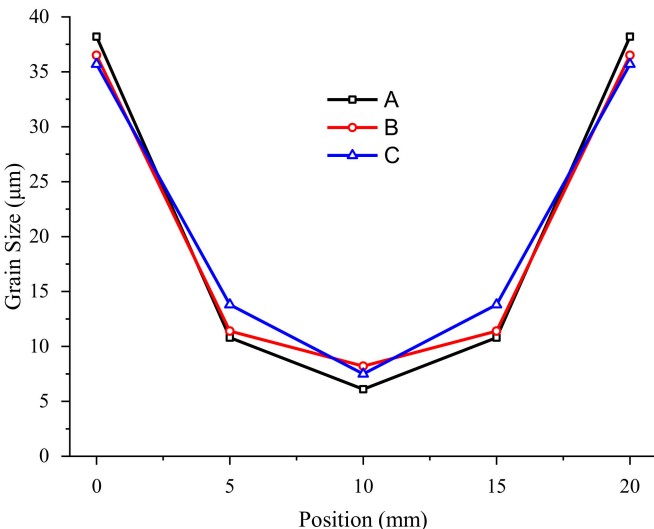

**Figure 17.** Grain size revolution regularity at specified regions.

The grain refinement during single-pass compression is mainly caused by the DRX, a particular mechanism of microstructure evolution. It is known as the only way to control the grain size of GH4169 in hot forming. It occurs at high temperatures when the deformation exceeds the critical strain, and the DRX volume percent increases with the rising temperature. The strain is much smaller on the surface of the specimen with the temperature below 1000 °C, where the DRX is unable to take place. Thus, large grains are present on the surfaces. The DRX behavior is more active when it occurs closer to the center, which promotes the grain refinement and results in a decrease in grain size. The grain structure in region B and region C presents great uniformity.

Grain sizes on the central plane (position 3) were taken into consideration. Three positions, A3, B3, and C3, share a similar average grain size of 5 μm~7 μm. The microstructures in Figure 16 show that the stretched origin grains are surrounded by many fine grains. Though the central materials hold high temperatures and large strain, the DRX occurs incompletely due to the short deformation time and immediate water cooling, which results in the coexistence of stretched grains and DRX grains.

Grain size reflects the material deformation temperature under the same strain rate, and meanwhile it reveals the plastic deformation characteristics. Both the temperature and plastic deformation are critical factors of forging RS. Comparing Figures 13 and 17, it can be concluded that there is a relationship between grain size and RS, as is seen in Figure 18. In the compressive regions, values of RS share the same trend of change as the grain size, while, in the tensile stress regions, the trends are contrary. Firstly, the DRX behavior is more active at high temperatures, and the grain size decreases while the grain amount increases, which results in the increase in material plasticity and deformation resistance. Since the temperature gradient is much greater during cooling, the tensile RS increase with the grain refinement. Secondly, the average grain size is greater at relatively low temperatures, and plastic deformation can more easily take place. The smaller temperature gradient induces lower tensile RS, and the compressive RS increase with the grain size.

The average grain size can be divided into 30 levels, where G00 represents the coarsest grain (over 508 μm) and G14.0 represents the minutest grain (below 2.8 μm). Thus, the RS level can be defined on the basis of the relationship mapping between RS and grain size. As is shown in Figure 19, the average grain size level of GH4169 after single-pass compression is G6.0~G11.0. The grain size level on the surfaces is G6.0, with the RS over 300 MPa. The grain size level on the center is G10.0 or even smaller, with the RS of 200 MPa~300 MPa. The statistical results show that, in the average grain size range of G7.0~G9.0, the variety of RS is approximately 100 MPa for every 1.0 change in grain size. Based on this conclusion, the RS level can be estimated according to the average grain size level, which is usually set as the standard requirement for forging. According to the residual stress and grain

distribution law of the blank, the optimal positions of the part in the blank ought to be arranged close to a quarter of the forging blank to guarantee low residual stresses and a homogeneous and refined grain.

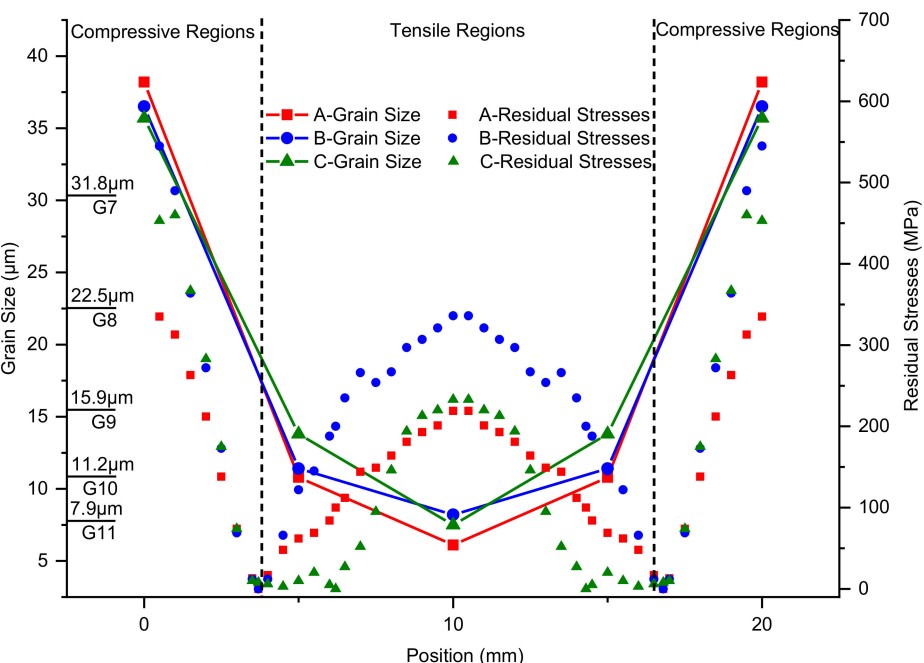

**Figure 18.** Mapping relationship between grain size and residual stresses.

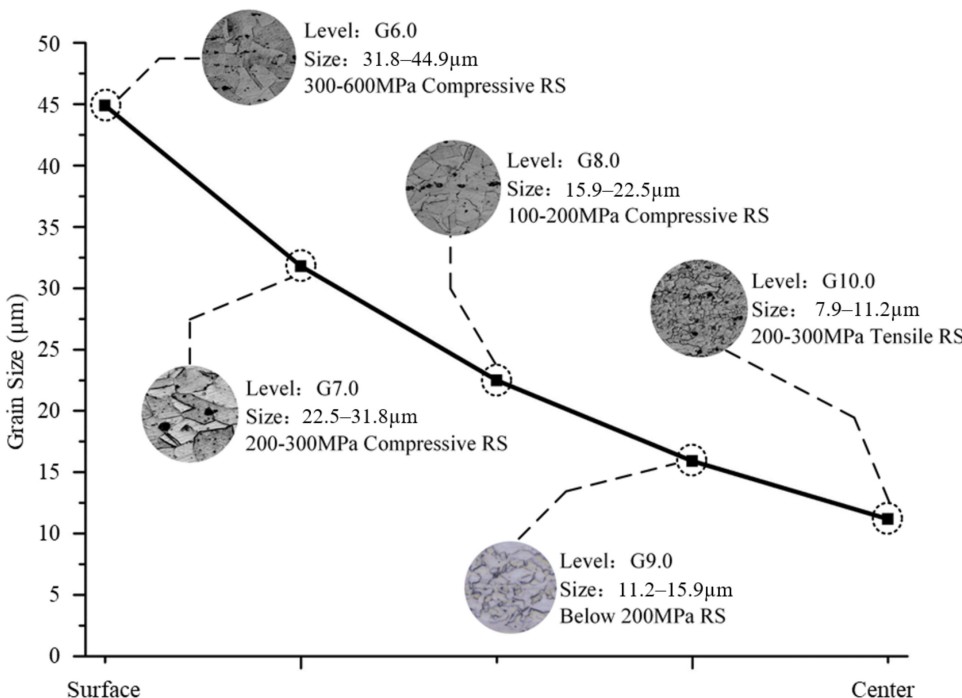

**Figure 19.** Residual stress division based on grain size.

## 5. Conclusions

The characterization of residual stresses and grain structure in the single-pass compression of GH4169 is introduced in this paper. The variations in residual stresses with forging temperature, loading speed, and cooling method are established by FE simulations. Experiments are conducted at a temperature of 1020 °C and loading speed of 25 mm/s,

with a press amount of 16 mm. This is immediately followed by water cooling. Based on the proposed new layer-stripping method, RS in the forging direction are tested and compared with the simulation results. The grain size evolution is summarized, and it is further used to realize RS level division. The main conclusions can be drawn as follows:

(1) The level of RS rises with the increase in forging temperature, loading speed, and cooling speed. Among them, loading speed mainly influences the stress value, and the cooling method has remarkable effects on both the value and distribution of RS. A small convective heat transfer coefficient and loading speed are recommended to obtain stable parts with uniform and low-level RS.

(2) A new layer-stripping method is put forward for interior RS measurement. The method is able to realize stress measurement in deep positions with high resolution. Measurement results are more accurate as they compensate for the stress redistribution caused by material removal. Compared with the traditional strain gauge layer-stripping method, the measurement efficiency of the new layer-stripping method is increased by 10 times.

(3) RS curves are nearly parabolic in depth, as the compressive RS change to tensile from the surfaces to the center. The transition layer with low-level RS should be retained in the subsequent machining. The peak values of compressive and tensile stresses appear on the surfaces and center, respectively. Three-fifths in the center are tensile layers, while the remainder shows compressive stresses. Nonuniform plastic deformation and temperature gradient are the two main factors that induce the RS during forging.

(4) Differences in grain size during single-pass compression are principally caused by the DRX. Grain size decreases from the surfaces to the center. Incomplete DRX results in the coexistence of stretched grains and refined grains. In the compressive regions, stress values share the same rules as grain size, while, in the tensile regions, they are contrary. In the average grain size range of G7.0~G9.0, the variety of RS is around 100 MPa for every 1.0 change in grain size. According to the residual stress and grain distribution law of the blank, the optimal positions of the part in the blank ought to be arranged close to a quarter of the forging blank to guarantee low residual stresses and a homogeneous and refined grain. Compared with the center position of the part in the blank, the residual stress of the part is reduced by 70%.

**Author Contributions:** Conceptualization, Z.W. and J.S.; methodology, Z.W. and J.G.; software, Y.Z.; validation, G.H., Z.W. and J.G.; formal analysis, Z.W.; investigation, Z.W.; resources, J.S.; data curation, Y.Z.; writing—original draft preparation, G.H., Y.Z. and Z.W.; writing—review and editing, Z.W., J.S., J.G., G.H. and W.C.; project administration, J.G. All authors have read and agreed to the published version of the manuscript.

**Funding:** This research was funded by the National Natural Science Foundation of China (grant number 51905021), the National Natural Science Foundation of China (grant number 51975034) and the Industry-University-Research Collaboration Project of Aero Engine Corporation of China (grant number HFZL2019CXY026).

**Institutional Review Board Statement:** Not applicable.

**Informed Consent Statement:** Not applicable.

**Data Availability Statement:** Not applicable.

**Conflicts of Interest:** The authors declare no conflict of interest.

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
