# Peer review of "Characterization of Residual Stresses and Grain Structure in Hot Forging of GH4169"

_aerospace, doi:10.3390/aerospace9020092_

Round 1

Reviewer 1 Report

The manuscript aerospace-1542839 addresses the forging process of GH4169. Therefore, cylindrical specimens under single pass compression were investigated. The authors applied an interesting combination of incremental hole drilling and layer removal to determine a deep residual stress profile. A finite element model is used to simulate the single pass compression process and to evaluate the effect of cooling speed, forging temperature, and loading speed. Residual stresses and grain size are analysed and recommendations of a part in a blank are given. My concerns are as follows:

Major concerns:

  • The proposed residual stress determination technique “laser-drilling & layer-stripping method” is not validated. Measurements of known residual stress profiles along the entire depth are needed to evaluate the accuracy of this method.
  • The numerical simulation is not validated by experiments. Only one simulation with the temperature of 1020℃ and loading speed of 25mm/s and water-cooling is compared to experiments.
  • Residual stress profiles and the grain size of microstructure before single pass compression is not given. Thus, changes of residual stresses and grain size caused by single pass compression are not clear. Residual stresses and grain size after cold-drawing and polishing have to be analysed as well.
  • Residual stresses of a blank are analysed. Residual stresses are in equilibrium and change when a part is cut from a blank. Thus, recommendations of optimal positions of the part in the blank based on the residual stress field of blank are questionable.

Detailed comments:

  • “Laser-drilling” is a confusing term and should be avoided. The applied method for residual stress determination is incremental hole drilling. Material is removed by drilling (not by a laser beam); surface deformation are measured by electronic speckle pattern interferometry.
  • Figure 2 indicates that the single pass compression process does not lead to homogeneous deformation (curved edges of the deformed area). Please explain how this is accounted for in the simulation.
  • Page 5, line 152: “Furthermore, measuring results will be corrected considering the stress redistribution caused by layer-stripping, which is always ignored in practical measmurements.”

This statement need a reference that proves that stress correction after layer removal during RS measurements is always ignored in literature, which I think it not true.

  • Page 5 line 154: “Surface RS within 2mm can be measured precisely every 0.2mm by the laser-drilling method and the first measured layer will then be removed.”

Please clarify the thickness of the first measurement layer. Is it 2 mm?

  • Page 5 line 166: “Symbol t represents the average thickness of each layer.”

Please clarify if different thickness layers were used in this work.

  • Figure 4: Please clarify if the half specimen is depicted (thickness of 10 mm). Layer n is at z=0 although the residual stress determination is intended to be applied only to one half of the specimen.
  • Page 7, line 195: “Compared with the traditional strain gauge layer-stripping method, the measurement efficiency of the laser-drilling & layer-stripping method is in-creased by 10 times.”

A reference is needed that gives the measurement accuracy of traditional methods. Additionally, a proper analysis of the accuracy of the proposed residual stress measurement technique is needed. Otherwise this statement has to be deleted.

  • Please clarify equation (3):

Page 5, line 169: “Symbols ??? and ??? respectively represent the RS in the directions of X and Y in layer i before removing,…”

Page 5, line 173: “Symbols ???′ and ???′ represent the net new balance stress of the rest materials after layer i being removed,…”

As layer 1 is removed it cannot contain residual stresses. Please clarify the position of stresses ???′ and ???′.

  • Commonly, the set-up of the numerical simulation is not part of the result section.
  • Please add the information of the used finite element software, the Young’s modulus and the poisons ratio of the material. Additionally, please add a figure showing the finite element mesh, mesh size, boundary conditions, and loadings of the finite element model.
  • Figure 7,8,9,10,12,14 need a coordinate system. I strongly recommend to use the same scale of respective subfigures to compare the results.
  • Page 9, line 267: “Consequently, the machine does more work to the specimen, which results in more temperature rise.”

Please clarify how heat generation is modelled in the finite element simulation.

  • Please explain why the simulation does not show an M-shaped residual stress profile at position D, although the simulation shows an M-shaped temperature profile.
  • Page 18, line 416: “Based on the conclusion, the RS level can be estimated according to the average grain size level, which is usually set as the standard requirements of forgings.”

Please explain the physical mechanism that connects residual stress and grain size. The grain size before single pass compression is not known, thus, the grain size could be a result of the cold-drawing and polishing processes.

  • I recommend to revise the English in the entire manuscript.

Reviewer 2 Report

The article is interesting, but a number of shortcomings need to be
corrected, for example:
1. The authors have to increase the font size in Fig. 4, 6, 15, 17, and 18.
2. It is necessary to describe in details the proposed method of determining residual stresses for better understanding of its unicity.
3. It is recommended to add more new References (2018-2021).

Round 2

Reviewer 1 Report

The authors addressed my comments.